# Generation of machine-learning derived cancer vulnerability indicator to determine the spatial burden of cancer outcomes

Kou Kou[1,2], Jessica Cameron[1,3,4], Paramita Dasgupta[1], Hao Chen[5], Peter D. Baade[1,4]*

1 Cancer Council Queensland, Brisbane, Australia, 2 School of Public Health and Social Work, Queensland University of Technology, Brisbane, Australia, 3 Faculty of Medicine, University of Queensland, Brisbane, Australia, 4 Centre for Data Science, Faculty of Science, Queensland University of Technology, Brisbane, Australia, 5 Australian Urban Research Infrastructure Network, Melbourne, Australia

* peter.baade@qut.edu.au

## Abstract

### Background

Due to the difficulty of obtaining population-based individual-level data, ecological studies are often used to explore factors related to geographic variations in health outcomes. This study proposes a novel framework to identify area-level predictors of spatial variations in lung cancer outcomes and generate a lung cancer vulnerability index (LcVI) based on these predictors.

### Methods

Data on 11,313 persons diagnosed with invasive lung cancer in Queensland, Australia (2016–2019) were sourced from the population-based Queensland Cancer Register. Bayesian spatial models estimated smoothed standardised incidence ratios (SIRs) for 519 geographic areas. Area-level variables (n = 911) were extracted from multiple data collections. Random forest models were fitted to identify important predictors for lung cancer incidence rates. A novel non-parametric dimensionality reduction approach incorporating the final random forest model results was developed to generate the LcVI which ranged from 0–10.

### Results

Eight variables were identified as predictors for lung cancer incidence with the top two being the prevalence of diabetes and adequate fruit intake. Areas having incidence rates below the Queensland average had significantly lower LcVI than those with average incidence rates (mean difference = 2.80, 95% CI: 2.34–3.25, p < 0.001) while areas with above average incidence rates had significantly higher LcVI than those with average incidence (mean difference = 2.70, 95% CI: 2.20–3.19, p < 0.001).

**Data availability statement:** The modelled area-level lung cancer incidence rates generated for this study are included in Supporting Information S1: Appendix 2; All area-level covariate data used in this study are publicly available from government open data sources. A complete list of dataset names and direct URLs is provided in Supporting Information S2 and are freely available alongside the published article.

**Funding:** The author(s) received no specific funding for this work.

**Competing interests:** The authors have declared that no competing interests exist.

The LcVI was strongly associated with the continuous SIR, explaining 57% of the variation ($R^2 = 0.57$, $p < 0.001$).

## Conclusion

This novel approach identified a small number of important predictors for lung cancer incidence from a high-dimensional dataset. The lung cancer vulnerability index partially explained the geographic variations, potentially offering insights into underlying drivers. As an ecological analysis, this associations reflect relationships at the population level. Future research incorporating individual-level data is needed to confirm whether the area-level associations observed here hold true for individuals.

---

## Introduction

With the expansion of cancer registries worldwide, we now have the ability to map geographic patterns of cancer outcomes [1,2], presenting a valuable opportunity to investigate the factors driving these variations. Since the data describing the potential drivers can be difficult to obtain at the individual level across the whole population, ecological studies using aggregated socio-environmental variables are commonly used to reveal important associations [3]. However, the vast amount of area-level data that are currently available poses challenges for traditional statistical models, which can struggle to manage the complexity of high-dimensional datasets.

Lung cancer was the most frequently diagnosed cancer and the leading cause of cancer-related deaths, globally responsible for almost 2.5 million new cases in 2022 [1]. Based on the Australian Cancer Atlas 2.0 [2], there was a four-fold disparity in small area-specific lung cancer incidence rates across Australia. Understanding the factors relating to this geographic disparity is crucial for the distribution of health resources and disease burden projection, especially as lung cancer is typically diagnosed at an advanced stage [4]. However, many studies that have examined the relationships between area-level socioeconomic indices and lung cancer incidence have reported inconsistent findings [1,5–7]. This inconsistency may be partly due to differences in the methodologies used to construct such indices [8], which are often generated using diverse socio-environmental variables without fully considering their direct relevance to cancer outcomes. Additionally, many dimensionality reduction methods that are commonly used for wide data sets, methods such as factor analysis or principal component analysis (PCA), combine correlated variables into artificial features, without assigning weights that reflect the specific associations of variables with cancer outcomes [9].

In this study, we developed a machine learning-based approach to identify important predictors of geographic patterns in cancer outcomes using routinely collected area-level data. Key predictors were then combined using a novel dimensionality reduction method, to generate a cancer vulnerability index (CVI) that helped explain spatial variations in cancer outcomes. This approach was applied to population-based lung cancer incidence data in Queensland, Australia.

## Methods

Approval was obtained from the data custodian to access de-identified routinely collected cancer incidence data from Queensland Cancer Register (QCR). The QCR is a legislated population-based registry. Information on all invasive cancers (excluding keratinocyte cancers) diagnosed in Queensland is required by law to be notified to the QCR [10]. From 2nd of September 2024, we got access to the QCR data for research purpose and the authors do not have access to information that could identify individual participants. Data quality is high, with 93% of cases histologically verified and 0.9% diagnosed by death certificate only in 2016 (unpublished data, QCR).

Outcome of interest (Fig 1, Orange boxes)

Records for all persons (n = 11,313) diagnosed with a primary invasive bronchial or lung cancer (ICD-10 code: C34.0-34.9) aged 20 or over in Queensland from 2016–2019 were extracted from the QCR. Cases without residential information (n = 39) and those with multiple primary lung cancer diagnoses (n = 67) were excluded, giving a final study cohort of 11,207 individuals. Patients' residential information at lung cancer diagnosis was defined by the 2016 Statistical Area Level 2 (SA2) [11]. In 2016 there were 528 SA2s covering Queensland without gaps or overlaps, of varying land area (median area 14 km$^2$, interquartile range (IQR): 6–95 km$^2$) and population (median: 7,857, IQR: 4,922–11,331) [11]. Nine SA2s with small populations (<5 residents annually, on average) were excluded from statistical modelling, leaving 519 SA2s for analysis.

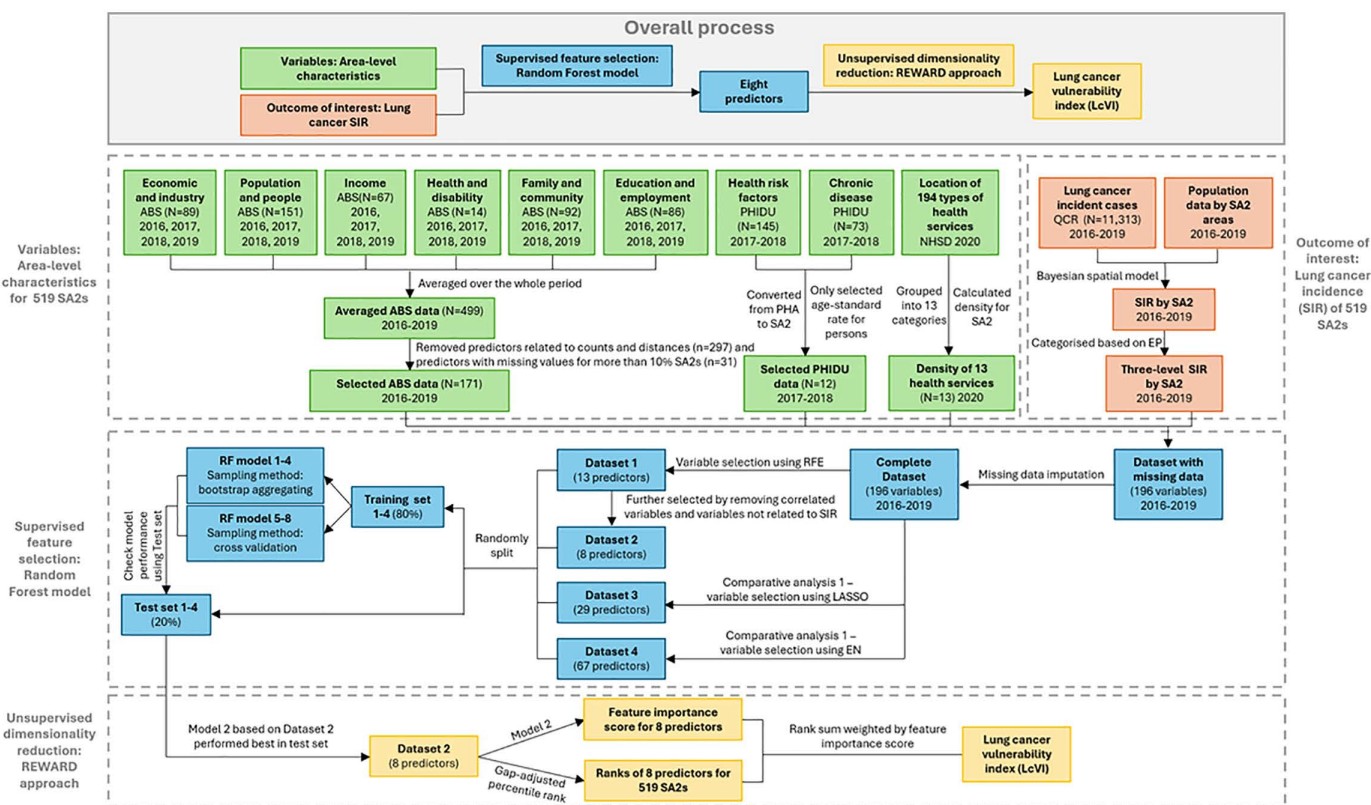

**Fig 1. Flowchart of generating machine learning derived lung cancer vulnerability index (LcVI).** Note: SIR: Standardised incidence ratio; ABS: Australian Bureau of Statistics; PHIDU: Public Health Information Development Unit; NHSD: National Health Services Directory; QCR: Queensland Cancer Register; PHA: Population Health Area; SA2: Statistical Area Level 2; SIR: Standardized incidence ratio; PP: Posterior probability; RFE: Recursive feature elimination; LASSO: Least absolute shrinkage and selection operator; EN: Elastic net; RF: Random forest.

The outcome of interest was the smoothed indirect standardized incidence ratios (SIR) of lung cancer during 2016–2019. SIRs reflect the relative incidence rate of lung cancer in each SA2 compared to the Queensland average, with high SIR values (>1) indicating that the incidence was above the Queensland average; SIR = 1 indicating similar incidence to the average; and low SIR values (<1) indicating below average incidence. The methods for estimating the SIRs using a Bayesian spatial model have been reported elsewhere [12,13] and further details are in Appendix 1.

In addition, the exceedance probability (EP) for each SA2 was estimated from the SIR model, and these provide statistical evidence indicating whether an area's incidence rates are greater than the state average [14]. Based on previous studies [15], we classified SA2s into three categories based on EP values, with low EP values (<0.2) providing evidence that incidence rates in those SA2s were below average (referred to as "below average"); values between 0.2 and 0.8 suggested a lack of evidence for a difference from the Queensland average ("average"); and high EP (>0.8) indicated the incidence rates were above average ("above average"). The smoothed SIR and EP for all the 519 SA2s in Queensland were listed in Appendix 2.

Population data for each SA2 by sex, calendar year and five-year age groups were obtained from the Australian Bureau of Statistics (ABS) [16].

Area-level characteristics (Fig 1, green boxes)

Area-level data from the ABS, Public Health Information Development Unit (PHIDU), and National Health Services Directory (NHSD) were retrieved from the Australian Urban Research Infrastructure Network (AURIN) Data Provider [17].

The ABS-sourced area level data for 2011–2019 [18] included 89 economic and industry [19], 151 population and people [20], 67 income [21], 14 health and disability [22], 92 family and community [23], and 86 education and employment variables [24]. For each SA2, data for each variable for at least one of the single years of 2016–2019 were extracted and averaged over the number of years the variable was available. Variables with missing average values for more than 10% of the 528 SA2s were excluded (n = 31). In addition, variables related to counts which are not comparable across SA2s were also excluded (n = 297), leaving 171 variables for analysis (Appendix 3).

The PHIDU-sourced [25] dataset included 145 variables relating to adult risk factors [26] and 73 variables relating to chronic disease in 2017–2018 [27]. Originally available at the level of Population Health Areas (PHA) which comprised a mix of single and aggregated SA2 areas, these were converted to SA2 via a population-weighed correspondence file [28]. Only variables reporting the age standardised rates (ASR) of each chronic disease and risk factor for persons were selected (n = 12, Appendix 3).

The NHSD [29] provided information on 194 types of health services by longitude and latitude coordinates, which were grouped into 13 categories (Appendix 4&5). In Queensland, as of November 2020, a total of 24,693 facilities fell within these 13 categories. For each SA2, the density per square kilometre for each category was determined based on the number of services within a buffered region of the SA2, determined by the SA2's geographical size and a 10-kilometer traversable accessibility network distance (as described in Appendix 6).

The final combined dataset included 196 area-level variables for each SA2 for 2016–2019.

Random Forest Weighted Gap Adjusted Percentile Rank Sum Index (REWIRED)

The REWIRED approach can be subdivided into two principal sections, a supervised variable selection phase (Fig 1, Blue boxes), and an unsupervised dimensionality reduction phase (Fig 1, Yellow boxes).

Supervised feature selection (Fig 1, Blue boxes)

The input data for supervised variable selection was the three-category outcome for lung cancer incidence ("below average", "average", "above average") and the 196 area-level variables for the 519 SA2s. This process incorporated missing data imputation, feature selection, model training, model performance evaluation, and feature importance estimation.

A random forest approach was used for imputation, and model building. Random forest is a non-parametric classification and regression tool for constructing prediction rules, that combines multiple tree predictors, with each tree trained on a sample randomly drawn from the original dataset using different sampling methods [30]. One advantage of random forests is that they do not make any prior assumptions on the form of their association with the outcome variable [30].

In the dataset, 106 out of 519 (20.4%) SA2s had at least one missing value among the 196 area-level variables. Missing data for the variables were imputed using a random forest model with the bagging (bootstrap aggregating) sampling method [31]. Bagging creates multiple bootstrap samples from the original dataset with replacement, and is robust against noisy data and outliers [32]. For missing data imputation, a decision tree is trained on each bootstrap sample using the remaining variables as predictors to predict the missing values. The final imputed value for each missing data point was obtained by averaging the predictions from all the trees. The random forest method has been demonstrated to significantly improve accuracy in missing data imputation compared to parametric imputation models [33]. After imputation, the 196 variables were centred by subtracting the mean value of each variable and scaled by dividing by its standard deviation.

We applied a non-parametric feature selection method, recursive feature elimination (RFE), to select predictor variables for the final random forest model [34]. RFE uses the random forest model to evaluate the importance of each predictor iteratively and eliminate the least important predictors in a backward manner. To increase the robustness of the final index [35], we set the RFE to select no more than 15 predictors, and this set of predictors was labelled the "RFE predictors". In addition, we identified a smaller subset of predictors by removing any predictors that had a correlation coefficient over 0.7 or less than −0.7 with other more important predictors; and any predictors that were not significantly associated with the continuous log SIR in linear regressions, and this set was labelled the "RFE_selected predictors". The model performances using these two sets of variables were compared using an 80% training set and 20% test set, ensuring that the distribution of the outcome variable was maintained in both sets.

Random forest involves several key hyperparameters including the number of decision trees in the forest; the minimal number of observation in a node (nodesize); and the number of predictors drawn randomly from the predictor list to include as candidate splitting predictors at each split (mtry) [36]. The number of trees is determined by the number of independent sub-samples drawn from the dataset, which depends on the sampling method. We compared two commonly used sampling methods in the model training process: bagging and K-fold cross-validation (CV). K-fold cross-validation is a method that divides the training dataset into K equal sets of observations, where each part is used once as validation data while the remaining K-1 parts are used for training [37,38]. This process is repeated K times, to train the random forest model, allowing for a more reliable estimate of its performance by averaging results across all K folds. To develop the final random forest models, we used bagging with 100 bootstrap samples for models 1 and 2; and 10-fold CV for model 3 and 4. We selected nodesize and mtry values based on the ones that yielded the highest Kappa, a model performance statistic (Appendix 7) [39].

The performance of the models was tested on the test dataset and accuracy was assessed using confusion matrices (Appendix 7) [40]. The model with the highest accuracy and Kappa value was chosen as the final model. Feature importance scores of the predictors were extracted from the results of this final model.

Unsupervised dimensionality reduction (Fig 1, Yellow boxes)

In this phase, the lung cancer vulnerability index (LcVI) was generated using the selected predictors and their feature importance scores via the REWIRED approach. This approach first rescaled the values for each predictor using the 'gap adjusted percentile rank' (described below), then weighted the rescaled values using their feature importance scores, and finally summed the weighted values for each area to create the LcVI.

The conventional percentile ranking method relies on the sequences of values, overlooking the absolute magnitude of differences between adjacent values. To address this limitation, we developed a 'gap-adjusted percentile rank,' accounting for varying intervals between values. This approach identified the minimum and maximum values of each continuous predictor, rescaling the predictor into 100 levels. Each observation was then assigned a rank based on its position within these levels, allowing for a standardized comparison within the predictor (Appendix 8). The gap-adjusted percentile rank ($p_{va}$) for a specific area ($a$) and predictor ($v_a$) can be calculated as:

$$p_{va} = \frac{\text{floor}\left[\frac{(v_a - v_{min})(n-1)}{v_{max} - v_{min}}\right] + 1}{n} \times 100$$

Where $v_{max}$ is the maximum value for variable v; $v_{min}$ is the minimum value for variable v; n is the number of observations; $\frac{v_{max}-v_{min}}{n-1}$ is the uniform gap between neighbouring ranks; $\text{floor}\left[\frac{(v_a-v_{min})(n-1)}{v_{max}-v_{min}}\right]+1$ is the gap adjusted rank ('floor' rounds down to the nearest integer; '+1' ensures ranks start at 1).

Following the 'gap-adjusted percentile rank' approach, predictor values were rescaled in ascending order. For predictors negatively associated with the outcome, values were inversely rescaled to ensure that the gap-adjusted percentile values for all predictors maintained a positive association with the outcome.

In contrast to the conventional percentile rank-sum approach, which merely aggregates percentile values of a set of variables, our approach also employed weighted summation to construct the LcVI. The weights of the predictors were the feature importance scores estimated from the final Random Forest model. The formulation is expressed as:

$$LcVI_a = \sum_v \frac{W_v p_{va}}{100}$$

Where $LcVI_a$ is the Lung cancer Vulnerability Indicator for area a; $W_v$ is the feature importance scores for predictor v; $p_{va}$ is the gap-adjusted percentile rank for predictor v in area a.

Extreme LcVI values below the 5th percentile and above the 95th percentile were truncated at these thresholds. Values were centred by subtracting the truncated minimum value and scaled by dividing the truncated range and then multiplying by 10 to get the final LcVI index values ranging from zero to ten. Hence zero represented the least vulnerable and ten the most vulnerable areas for the diagnosis of lung cancer.

An ANOVA test followed by a post-hoc analysis using Tukey's Honest Significant Difference (HSD) test, a linear regression model, and various visualisations were used to evaluate the capacity of the LcVI to explain the geographic variation of lung cancer incidence in Queensland.

## Comparative analysis

Parametric variable selection methods were also used to select predictors and compare estimated models with the RFE models (Fig 1, Blue boxes). We applied Least Absolute Shrinkage and Selection Operator (LASSO) regression and Elastic Net (EN) regression as the parametric feature selection methods [41,42] (Appendix 9). The performance of models 5–8 based on features selected using LASSO and EN were compared with models 1–4 (Table 1).

In addition, the performance of LcVI in predicting the SIR of lung cancer was compared with the performance of the Australian area disadvantage index, the Relative Socio-economic Disadvantage (IRSD) index [43]. The IRSD index is a general socio-economic index that summarises a range of information about the economic and social conditions of people and households within an area. It ranges from one to ten, with a score of one representing the least disadvantaged areas and ten representing the most disadvantaged areas. The same statistical tests and visualisations were also used to evaluate the capacity of the IRSD to explain the geographic variation of lung cancer incidence in Queensland.

All analyses were performed with R version 4.4.0. Bayesian spatial model was fitted using the 'CARBayes' package (version 6.1.1) and random forest models used the 'caret' package (version 6.0.94). Appendix 10 gives an example of syntax showing the key functions used in the REWIRED framework.

## Results

Between 2016 and 2019, the SIR of lung cancer for the 519 SA2s ranged from 0.46 to 1.60. There was evidence that lung cancer incidence rates were below the Queensland average in 158 (30.4%) SA2s, above the average in 120 (23.2%) SA2s, and not different from the average in the remaining 241 (46.4%) SA2s.

Eight random forest models were developed based on different variable selection and sampling methods (Table 1). Model 2 based on eight predictors selected using the RFE method (RFE_selected predictors) and bagging sampling

**Table 1. Model performance based on test set.**

| Model[1] | No. predictors[2] | Overall | | Average | | Below average | | Above average | |
|---|---|---|---|---|---|---|---|---|---|
| | | Accuracy[3] | Kappa[4] | Sensitivity[5] | Specificity[6] | Sensitivity | Specificity | Sensitivity | Specificity |
| 1. boot_RFE | 13 | 0.61 | 0.39 | 0.60 | 0.64 | 0.74 | 0.85 | 0.46 | 0.89 |
| **2. boot_RFE_selected** | **8** | **0.63** | **0.42** | **0.60** | **0.67** | **0.74** | **0.85** | **0.54** | **0.89** |
| 3. CV_RFE | 13 | 0.60 | 0.38 | 0.52 | 0.69 | 0.77 | 0.81 | 0.54 | 0.87 |
| 4. CV_RFE_selected | 8 | 0.56 | 0.32 | 0.52 | 0.62 | 0.64 | 0.82 | 0.54 | 0.86 |
| 5. boot_LASSO | 29 | 0.61 | 0.39 | 0.56 | 0.67 | 0.77 | 0.83 | 0.50 | 0.87 |
| 6. boot_EN | 67 | 0.58 | 0.34 | 0.56 | 0.62 | 0.74 | 0.82 | 0.42 | 0.89 |
| 7. CV_LASSO | 29 | 0.58 | 0.35 | 0.52 | 0.65 | 0.81 | 0.82 | 0.42 | 0.86 |
| 8. CV_EN | 67 | 0.55 | 0.31 | 0.48 | 0.64 | 0.77 | 0.79 | 0.42 | 0.86 |

Note: 1.'boot' refers to models using bootstrap aggregating as the sampling method; 'CV' refers to models using cross-validation as the sampling method;'RFE' refers to features selected using the recursive feature elimination approach;'RFE_selected' refers to the RFE features which were further selected by removing correlated features and features not related to SIR; LASSO' refers to features selected using the least absolute shrinkage and selection operator approach; 'EN' refers to features selected using the elastic net approach.

2. Lists of predictors for model 1–4 were presented in Appendix 11.

3. Accuracy refers to the proportion of predictions that were correctly predicted.

4. Kappa is a measure of agreement between actual and predicted classes, considering the agreement occurring by chance.

5. Sensitivity refers to the proportion of positives that were true positives.

6. Specificity refers to the proportion of negatives that were true negatives.

method achieved the highest Kappa statistics (Kappa = 0.42) and accuracy (accuracy = 0.63, 95% CI: 0.53–0.72, no information rate = 0.47, p < 0.001) in the test set (See Appendix 7&12 for more information). The model correctly identified 74% of the 'below average' SA2s, 54% of the 'above average' SA2s, and 60% of the 'average' SA2s, indicating its sensitivity for each category. In terms of specificity, the model correctly identified 85%, 89%, and 67% of areas as not belonging to the 'below average,' 'above average,' and 'average' categories, respectively.

Table 2 lists the eight RFE_selected predictors with their feature important scores generated by Model 2. The prevalence of diabetes had the highest feature importance score. SA2s categorised as 'below average' had a median age-standardised prevalence rate (ASR) of diabetes as 3.78, contrasting with a median ASR of 6.07 for SA2s with 'above average' lung cancer incidence. The univariate linear regression model showed that this variable alone explained 45% of the lung cancer geographic variation ($R^2 = 0.45$, Table 2).

Based on the median values, the prevalence rates of diabetes, being unemployed, lower education level, and a marital status of 'separated' were positively associated with lung cancer incidence; while the prevalence rates of adequate fruit intake, having private health insurance and higher education, and the median superannuation and annuity income were negatively associated with lung cancer incidence.

Areas with 'below average' SIR had the lowest scores of the generated index (LcVI) (median = 2.28, Q1-Q3: 0.85–3.72) compared to SA2s with 'average' SIR (median LcVI = 5.30, Q1-Q3: 3.88–6.55) and 'above average' SIR (median LcVI = 8.03, Q1-Q3: 6.98–9.16) (Fig 2). The ANOVA test showed significant differences in the LcVI values across the three categories of SIR ($F_{(2, 516)} = 287.8$, p < 0.001). Post-hoc analysis further suggested a clear gradient in the LcVI by SIR, with the 'below average' category having a significantly lower LcVI compared to the 'average' (mean difference = 2.80, 95% CI: 2.34–3.25, p < 0.001); while the 'above' category had a significantly higher LcVI than the 'average' (mean difference = 2.70, 95% CI: 2.20–3.19, p < 0.001).

The LcVI was significantly associated with the continuous SIR for the 519 SA2s, with a Peason correlation of 0.76. On linear regression, the LcVI explained 57% of the variation in the (natural) logarithm-transformed SIR across Queensland, with each one-point increase in LcVI associated with a 6% increase in SIR ($R^2 = 0.57$, coef. = 0.06, t = 26.3, p < 0.001). In

**Table 2. Feature importance scores and median values[1] by categories of lung cancer incidence for the selected predictors.**

| Features | Feature importance score | Median values by category | | | R square[3] |
|---|---|---|---|---|---|
| | | Below | | Average | Above |
| Diabetes mellitus (ASR[2]) | 81.55 | 3.78 | 4.77 | 6.07 | 0.45 |
| Adequate fruit intake (ASR) | 58.35 | 52.71 | 50.52 | 48.59 | 0.33 |
| High-income taxpayers with private health insurance (%) | 55.69 | 56.49 | 46.45 | 33.09 | 0.37 |
| Bachelor's degree (%) | 48.19 | 18.25 | 10.30 | 7.25 | 0.43 |
| Unemployed (%) | 44.52 | 5.90 | 6.90 | 10.00 | 0.18 |
| Median annual income-Superannuation and annuity ($1000) | 34.58 | 20.52 | 17.27 | 15.09 | 0.24 |
| Highest year of school completed-Year 11 or equivalent (%) | 32.63 | 6.60 | 8.10 | 8.60 | 0.28 |
| Marital status-Separated (%) | 29.73 | 2.90 | 3.60 | 4.35 | 0.29 |

Note: 1. Values were standardised when included in random forest models; 2. ASR: age standardised prevalence rate per 100 people. 3. $R^2$ is based on linear regression between continuous log2(SIR) and the specific feature.

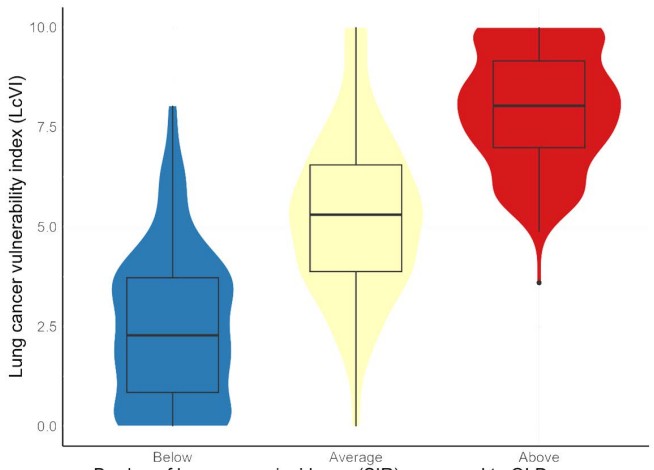

**Fig 2. Distribution of the lung cancer vulnerability index (LcVI) by categories of lung cancer standardised incidence ratio (SIR) across Queensland.** *Note: The violin plots show the distribution of LcVI values for areas classified as having below-average (blue), average (yellow), or above-average (red) lung cancer incidence. The LcVI ranges from 0 to 10, with higher values indicating greater area-level vulnerability. Areas with below-average SIR had a median LcVI of 2.28 (Q1–Q3: 0.85–3.72), those with average SIR had a median of 5.30 (Q1–Q3: 3.88–6.55), and those with above-average SIR had a median of 8.03 (Q1–Q3: 6.98–9.16). The clear gradient across categories demonstrates that the LcVI effectively differentiates areas by their observed lung cancer incidence.*

the top graph of Fig 3, the triangles and dots for each column represent the LcVI and SIR of the same SA2, with colour representing the value of SIR. Most of the SA2s with low SIR (blue) had low LcVI (triangles down in the bottom), and vice versa. The comparative analysis showed that, compared to the IRSD index, the LcVI explained more variation in SIR across Queensland (42% vs 57%). The lower graph of Fig 3 also shows a less significant association between SIR and IRSD.

## Discussion

Lung cancer is one of the most preventable cancers, with over 80% of cases attributable to tobacco smoking and other environmental predictors [44]. However, due to the significant latency period between exposure to these risk factors and lung cancer diagnosis [45], it is difficult to establish a direct relationship between the prevalence of risk

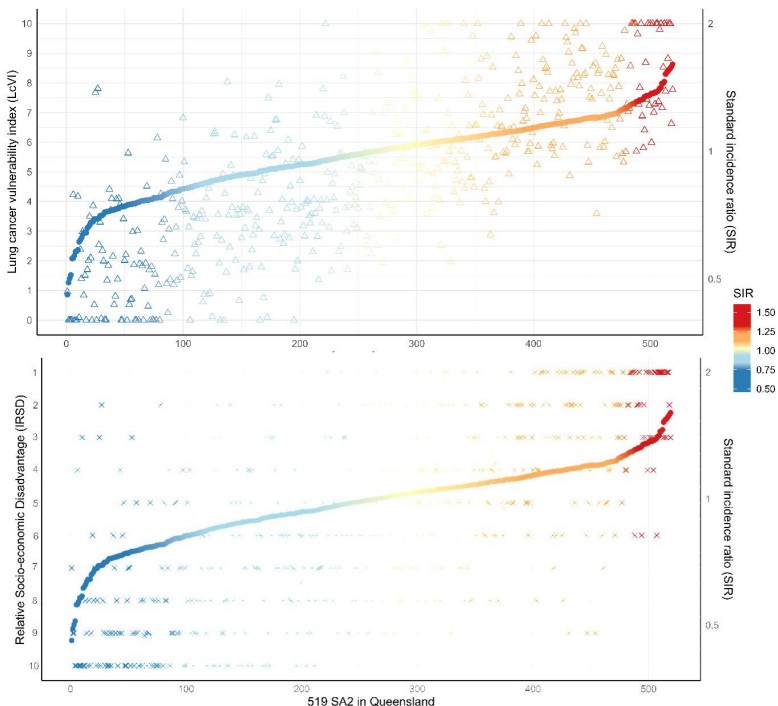

**Fig 3. Relationship between lung cancer vulnerability index (LcVI, upper) or relative socio-economic disadvantage (IRSD, lower) and lung cancer standardised incidence ratio (SIR) across 519 Statistical Areas Level 2 (SA2s) in Queensland.** *Note: Each column represents an SA2, with triangles showing the LcVI values (upper) and crosses showing the IRSD values (lower), dots showing the SIR values of lung cancer. Areas with higher LcVI values generally correspond to areas with higher SIR, demonstrating a strong positive association between the index and observed incidence. For comparison, the lower panel presents the relationship between the area disadvantage (IRSD) and SIR, where the y-axis for IRSD has been reversed to align the direction of the trend with that of the LcVI. The consistent colour gradient across both panels (blue to red) reflects increasing lung cancer incidence (SIR) from below- to above-average areas.*

factors and lung cancer incidence in ecological studies. While not specifically overcoming this limitation, this study proposes a standard framework, the REWIRED approach, to identify important area-level characteristics that might serve as a proxy to reflect the historical prevalence of lung cancer predictors, thus providing further understanding of reasons for the geographic variations of lung cancer incidence. Based on eight area-level characteristics, the final model achieved high specificity (0.89) for predicting areas with above-average incidence. This high specificity indicates that if the model predicts an area as not above average, it is highly likely that the lung cancer incidence in that area is indeed not above average. Similarly, the model performs well in predicting areas with below-average incidence, achieving a specificity of 0.85, meaning it effectively avoids falsely identifying areas as below average when they are not. The lung cancer specific index generated based on the eight predictors explained more (57% vs 42%) of the geographic variation compared to the more generic area-level socioeconomic disadvantage index based on 16 characteristics [43].

There are several advantages of using the REWIRED approach. Firstly, this approach is flexible with different data patterns by using non-parametric techniques in both supervised variable selection (Random Forest models) and dimensional reduction (gap-adjusted percentile rank-sum) processes. Compared with regression models for feature selection and PCA for dimensional reduction, this non-parametric approach is able to address non-linear data, which is a typical characteristic of area-level datasets. Comparative analysis also showed that the non-parametric variable selection using RFE performed better than the parametric variable selection methods using LASSO or EN methods.

In contrast to common unsupervised dimensional reduction approaches such as PCA and percentile rank sum which treat all variables equally without considering their relationship with health outcomes, the REWIRED approach weighted the predictors based on the feature importance scores derived from Random Forest models, which allowed for the construction of a vulnerability index (LcVI) that reflects the relative importance of each variable.

It is important to note that the feature importance scores derived from the random forest model reflect the predictive contribution of each area-level variable to the spatial pattern of lung cancer incidence, rather than causal strength. Consequently, the weights applied in constructing the LcVI are grounded in predictive relevance, not causality. This aligns with the ecological nature of the study, which aims to identify spatially correlated characteristics rather than establish cause–effect relationships. The strong association between the LcVI and observed incidence ($R^2 = 0.57$) supports the index's capacity to capture meaningful spatial variation, while also providing a foundation for future studies to test these associations at the individual level.

To our knowledge, our study is the first ecological study to report a link between the population prevalence of diabetes and the incidence of lung cancer. The age-standardized prevalence rate of diabetes achieved the highest feature importance during the REWIRED process, explaining 45% of the geographic variation in lung cancer incidence. Previous studies have linked increased diabetes risk with various smoking behaviours, including active smoking, passive smoking, and smoking cessation, showing a dose-response relationship between smoking and diabetes risk [46]. Therefore, while our results would need confirmation through other studies, the population prevalence of diabetes could potentially provide a more effective indicator of long-term smoking behaviours among the residents compared to the current smoking prevalence, although this should be interpreted as a hypothesis that requires further empirical validation. In addition, since diabetes usually causes symptoms and requires treatment, population-level estimates of diabetes prevalence could be more reliable and accessible than corresponding estimates of smoking prevalence. Our findings showed a high correlation between the prevalences of diabetes and current smoking at the small area level (Pearson's correlation coefficient = 0.78). Of note, the predictor relating to current smoking prevalence had a lower feature importance score in predicting lung cancer incidence so was excluded from the final model. This could be expected given the often 20–30 year lag period between smoking patterns and lung cancer incidence patterns [47]. If this interpretation is confirmed through further validation, this may increase the rationale for using diabetes prevalence as an ecological measure of previous smoking behaviour, rather than current smoking prevalence.

The prevalence of adequate fruit intake was identified as the second most important feature related to the spatial variation of lung cancer incidence in our study. Inadequate fruit intake has previously been identified as having a causal association with lung cancer, and is an attributable factor in as many as 9.6% of lung cancer diagnoses [44,48]. In addition, a meta-analysis of 38 individual-level studies also reported a significant protective effect of adequate fruit intake against lung cancer [49]. On the area level, a more plausible explanation for our results may be that adequate fruit intake is a proxy for broader health-conscious behaviours, as individuals who meet dietary guidelines for fruit consumption are likely to engage in other healthy lifestyle practices [50]. Additionally, the Australian Cancer Atlas 2.0 indicates similarity between those areas that have a higher prevalence of health predictors such as smoking and those areas that have higher rates of inadequate food intake, further supporting the suggested link between dietary habits and overall health behaviours [2].

This study compared the LcVI with the IRSD index as a reference point, recognising that the IRSD reflects general socioeconomic disadvantage rather than cancer-specific vulnerability. Many previous studies have used indices representing general socioeconomic disadvantage to examine associations with lung cancer outcomes [1,5–7]; therefore, this comparison was intended to demonstrate whether a lung cancer-specific index could provide additional explanatory power beyond commonly used socioeconomic measures. Although a few studies have developed cancer-related vulnerability indices for specific cancers such as skin cancer [51] or for behavioural predictors influencing cancer risk [52], there is currently no widely adopted or externally validated index specifically designed to quantify area-level vulnerability to lung cancer incidence. The LcVI thus represents one of the first attempts to construct a lung cancer-specific vulnerability index directly linked to spatial

variations in lung cancer risk. Nevertheless, the LcVI has not yet been externally validated, and its generalisability to other settings remains to be confirmed. Future research should test the reproducibility of this framework using independent datasets and explore its adaptability for other cancers or geographic contexts. Some of the key features identified in this study overlap with those used in generating the IRSD index [43]. Whether these features are individually important in explaining the variation in lung cancer diagnosis, or collectively measure some aspect of socioeconomic status, remains uncertain.

As an ecological analysis, this study is subject to the inherent limitation of ecological fallacy. The associations identified between area-level characteristics and lung cancer incidence reflect relationships at the population or geographic level and cannot be assumed to apply to individuals within those areas. For example, areas with higher diabetes prevalence or lower fruit intake may have higher lung cancer incidence, but this does not imply that individuals with diabetes or inadequate fruit intake necessarily have an increased risk of lung cancer. The purpose of this study was to identify spatial patterns and potential contextual determinants of disease burden, not to infer individual-level causation. Future research incorporating individual-level data is needed to confirm whether the area-level associations observed here hold true for individuals. Another limitation of this study is the absence of small-area historical data on smoking behaviour or diabetes prevalence, which prevented the incorporation of temporal lagged variables.

Despite these limitations, the strong association between the LcVI and lung cancer incidence demonstrates that those area-level characteristics identified in this study are powerful predictors of the geographic variation in lung cancer burden. Such evidence could inform targeted prevention, screening, and resource allocation strategies. In addition, by comparing the LcVI with existing cancer surveillance data, public health agencies can identify outlier regions – such as areas with low LcVI (favourable area characteristics) but unexpectedly high lung cancer incidence – suggesting the presence of other unmeasured risk factors that warrant further investigation. Finally, the REWIRED framework underlying the LcVI can be adapted to other cancer types, enabling the development of comparable vulnerability indices that support comprehensive, data-driven cancer control strategies across different diseases and geographic contexts. Conclusion

This study developed a novel framework to conduct robust ecological analyses of lung cancer outcomes, with the potential to extend this to other cancer types and outcomes. The REWIRED approach showed better explanatory power for geographic variations in lung cancer incidence compared to standard population-based area-level socioeconomic indices. While these area-level results cannot be extrapolated to the individual level, by using easily accessible ecological data they provide important insights into the underlying drivers of the observed geographic disparities in lung cancer incidence and so have an important role in guiding further research. As this is an ecological study, the findings should be interpreted with caution due to the potential for ecological fallacy.

## Supporting information

**S1 Appendice. Appendices 1–12.**
(DOCX)

**S2 File. List of area-level datasets and direct URLs.**
(DOCX)

## Acknowledgments

The authors express their gratitude to the Data Custodian, as well as the staff of the Australian Urban Research Infrastructure Network (AURIN) and the Queensland Cancer Register (QCR), for their support in providing the relevant datasets.

## Author contributions

**Conceptualization:** Kou Kou, Peter Baade.

**Data curation:** Kou Kou.

**Formal analysis:** Kou Kou.

**Funding acquisition:** Peter Baade.

**Investigation:** Kou Kou.

**Methodology:** Kou Kou, Jessica Cameron, Hao Chen.

**Project administration:** Peter Baade.

**Resources:** Peter Baade.

**Software:** Kou Kou, Peter Baade.

**Supervision:** Peter Baade.

**Validation:** Kou Kou.

**Visualization:** Kou Kou.

**Writing – original draft:** Kou Kou.

**Writing – review & editing:** Kou Kou, Jessica Cameron, Paramita Dasgupta, Hao Chen, Peter Baade.

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
