## [Decision Letter · Decision Letter 0]

13 Oct 2025

Dear Dr.  Baade,

We look forward to receiving your revised manuscript.

Kind regards,

Godwin Banafo Akrong, Ph.D.

Academic Editor

PLOS ONE

Journal Requirements:

3. In the online submission form, you indicated that “The links for area-level variables have been provided in references. Modelled area-level lung cancer incidence rates are available by contacting the corresponding author.”

Additional Editor Comments:

Key Areas for Improvement:

Validate whether these weights meaningfully reflect causal associations or merely statistical correlations (Reviewer #1)The comparison between the LcVI and the IRSD index is a key area for improvement. (Reviewer #1 and Reviewer #3)Reviewer #1 suggests further improvements to the study's conclusion.Interpretation of diabetes prevalence (Reviewer #3)Clarify the risk of ecological inference as suggested by Reviewer #3

I encourage you to address all the reviewers' (#1 and #3) comments and make the necessary revisions.

Reviewer's Responses to Questions

**Comments to the Author**

1. Is the manuscript technically sound, and do the data support the conclusions?

Reviewer #1: Yes

Reviewer #3: Yes

2. Has the statistical analysis been performed appropriately and rigorously?

Reviewer #1: Yes

Reviewer #3: Yes

3. Have the authors made all data underlying the findings in their manuscript fully available?

Reviewer #1: No

Reviewer #3: Yes

4. Is the manuscript presented in an intelligible fashion and written in standard English?

Reviewer #1: Yes

Reviewer #3: Yes

Reviewer #1: This is a study focusing on machine-learning derived cancer vulnerability indicator to determine the spatial burden of cancer outcomes. Unfortunately, however, this paper needs further optimization of the results and statistical analysis.

1.The feature importance scores from the random forest model are used to weight the vulnerability index, yet the paper does not validate whether these weights meaningfully reflect causal associations or merely statistical correlations, potentially introducing bias into the index construction.

2. The study does not explicitly incorporate temporal lagged variables (e.g., historical smoking data or longitudinal diabetes trends) to test this hypothesis.

3.The comparison of LcVI with the IRSD index is overly simplistic, as IRSD is a general socioeconomic disadvantage measure, not a cancer-specific index.

4.The conclusion suggests the framework can be extended to other cancer types, but the study exclusively analyzes lung cancer in a single region.

Reviewer #3: The manuscript introduces a novel methodological framework (REWIRED) for generating a Lung Cancer Vulnerability Index (LcVI) using ecological, registry-based, and socio-environmental data in Queensland, Australia. The approach is technically strong, innovative, and relevant to cancer epidemiology and public health. The paper is generally well written and makes a meaningful contribution.

Overall, the work is sound and the results are convincing. My comments mainly concern interpretation and clarity, and I believe these can be addressed with relatively minor revisions.

Main Comments:

1. Interpretation of diabetes prevalence: The finding that diabetes prevalence was the strongest predictor is interesting. However, the claim that it may serve as a proxy for historical smoking behaviour should be framed more cautiously as a hypothesis, unless further evidence is provided.

2. Ecological fallacy: The risk of ecological inference should be emphasized more clearly throughout, to ensure readers do not mistakenly extend conclusions to individual-level risk.

3. Validation: The LcVI is validated internally but not externally. Please expand discussion of this limitation and its implications for generalizability.

4. Clarity of presentation: Some sections describing the REWIRED method are mathematically dense. A simplified schematic or clearer narrative would help readers unfamiliar with advanced ML methods.

Minor Points:

1. Figures 2 and 3 would benefit from more descriptive captions.

2. The abstract wording (e.g., “effectively explained”) could be tempered slightly to avoid overstating results.

3. Use consistent terminology (“predictors” vs. “risk factors”).

4. Strengthen the discussion of how public health agencies might use the LcVI in practice.

Data Availability:

Most area-level datasets are openly available. However, the core registry data are subject to custodian approval, which should be clearly acknowledged in the data availability statement.

**Do you want your identity to be public for this peer review?** For information about this choice, including consent withdrawal, please see our Privacy Policy

Reviewer #1: No

Reviewer #3: No

---

## [Author Response · Author response to Decision Letter 1]

9 Dec 2025

Additional Editor Comments:

Key Areas for Improvement:

1. Validate whether these weights meaningfully reflect causal associations or merely statistical correlations (Reviewer #1)

2. The comparison between the LcVI and the IRSD index is a key area for improvement. (Reviewer #1 and Reviewer #3)

3. Reviewer #1 suggests further improvements to the study's conclusion.

4. Interpretation of diabetes prevalence (Reviewer #3)

5. Clarify the risk of ecological inference as suggested by Reviewer #3

I encourage you to address all the reviewers' (#1 and #3) comments and make the necessary revisions.

Response to Editor:

We sincerely thank the Editor for summarising the key areas for improvement and for the constructive guidance. We have carefully revised the manuscript in accordance with these points, as well as the detailed comments from Reviewers #1 and #3. The main changes are summarised below:

1.Validation of feature weights (Reviewer #1):

We clarified in the Discussion that the feature importance scores derived from the random forest model reflect predictive strength rather than causal associations. We emphasised that the LcVI was designed to capture predictive relationships at the area level, not to infer causality. A new paragraph was added (details below) explaining that these weights quantify predictive relevance, supported by the strong association between the LcVI and observed lung cancer incidence (R² = 0.57, p < 0.001).

2.Comparison between LcVI and IRSD (Reviewers #1 and #3):

We revised the Discussion to better justify the use of the IRSD index as the most appropriate reference for comparison, acknowledging that IRSD represents general socioeconomic disadvantage rather than cancer-specific vulnerability. The changes are detailed below, in response to the Reviewers’ questions. Briefly, we added that few cancer-related indices (e.g., for skin cancer or behavioural predictors) exist, and none are widely validated or lung cancer-specific at the small-area level. We also highlighted the need for external validation of the LcVI and outlined future directions for testing its reproducibility and generalisability.

3. Improvement of the conclusion (Reviewer #1):

The conclusion has been strengthened to emphasise the study’s contribution and future applications. We now highlight that the LcVI effectively captures spatial variation in lung cancer incidence, provides a framework for hypothesis generation, and can guide future research and targeted interventions.

4. Interpretation of diabetes prevalence (Reviewer #3):

We revised the relevant section in the Discussion to frame the interpretation more cautiously, clarifying that the hypothesis that diabetes prevalence may act as a proxy for historical smoking behaviour requires further empirical validation. The added text states that this interpretation should be viewed as preliminary and subject to confirmation in future studies.

5. Clarification of ecological inference (Reviewer #3):

A new paragraph was added at the end of the Discussion addressing the limitation of ecological fallacy. We clarified that all associations reported are at the area level and should not be inferred to individuals. The paragraph also explains that this study identifies spatial patterns and contextual determinants, while individual-level studies are needed to confirm causality.

We believe these revisions have substantially improved the clarity, interpretability, and practical relevance of the manuscript. We thank the Editor and Reviewers again for their insightful feedback, which has strengthened the paper considerably.

Reviewer #1: This is a study focusing on machine-learning derived cancer vulnerability indicator to determine the spatial burden of cancer outcomes. Unfortunately, however, this paper needs further optimization of the results and statistical analysis.

1.The feature importance scores from the random forest model are used to weight the vulnerability index, yet the paper does not validate whether these weights meaningfully reflect causal associations or merely statistical correlations, potentially introducing bias into the index construction.

Response: This study is ecological in design and so cannot infer causal relationships. Instead, it aims to identify area-level characteristics that are associated with geographic variations in lung cancer incidence. The feature importance scores from the random forest model reflect the predictive strength of each variable for distinguishing areas with different incidence levels. Features with higher scores contribute more to the predictive capacity of the model, and thus were assigned higher weights in constructing the lung cancer vulnerability index (LcVI).

The LcVI was designed to assess whether the identified predictors collectively explain spatial variation in incidence. This was validated through a strong association between LcVI and lung cancer incidence (R² = 0.57, p < 0.001), indicating that the weighted index effectively captures meaningful area-level differences related to disease burden. Therefore, the weighting scheme is justified within the predictive framework of this ecological analysis, even though causal inference is beyond the study’s scope.

To make this clearer, we have added a statement in the Discussion highlighting that the feature weights are derived from predictive importance rather than causal strength, and that the results provide a foundation for hypothesis generation and further individual-level research:

“It is important to note that the feature importance scores derived from the random forest model reflect the predictive contribution of each area-level variable to the spatial pattern of lung cancer incidence, rather than causal strength. Consequently, the weights applied in constructing the LcVI are grounded in predictive relevance, not causality. This aligns with the ecological nature of the study, which aims to identify spatially correlated characteristics rather than establish cause–effect relationships. The strong association between the LcVI and observed incidence (R² = 0.57) supports the index’s capacity to capture meaningful spatial variation, while also providing a foundation for future studies to test these associations at the individual level.”

2. The study does not explicitly incorporate temporal lagged variables (e.g., historical smoking data or longitudinal diabetes trends) to test this hypothesis.

Response:

Due to the lack of historical data on smoking behaviour or diabetes prevalence at the small-area level, temporal lagged variables could not be incorporated into this analysis. We have added a statement in the Discussion to acknowledge this data limitation as:

“Another limitation of this study is the absence of small-area historical data on smoking behaviour or diabetes prevalence, which prevented the incorporation of temporal lagged variables.”

3.The comparison of LcVI with the IRSD index is overly simplistic, as IRSD is a general socioeconomic disadvantage measure, not a cancer-specific index.

Response: We agree that the IRSD index is a poor measure of cancer vulnerability. However, as no established lung cancer-specific index currently exists, the IRSD was chosen as the most relevant available benchmark for contextual comparison within Queensland. It provides a well-recognised reference for assessing whether our cancer-specific index offers improved explanatory power over standard socioeconomic measures commonly used in previous ecological cancer studies.

We have added a statement in the Discussion section to clarify these points and acknowledge this limitation:

“This study compared the LcVI with the IRSD index as a reference point, recognising that the IRSD reflects general socioeconomic disadvantage rather than cancer-specific vulnerability. Many previous studies have used indices representing general socioeconomic disadvantage to examine associations with lung cancer outcomes [1-4]; therefore, this comparison was intended to demonstrate whether a lung cancer-specific index could provide additional explanatory power beyond commonly used socioeconomic measures. Although a few studies have developed cancer-related vulnerability indices for specific cancers such as skin cancer [5] or for behavioural risk factors influencing cancer risk [6], there is currently no widely adopted or externally validated index specifically designed to quantify area-level vulnerability to lung cancer incidence. The LcVI thus represents one of the first attempts to construct a lung cancer-specific vulnerability index directly linked to spatial variations in lung cancer risk. Nevertheless, the LcVI has not yet been externally validated, and its generalisability to other settings remains to be confirmed. Future research should test the reproducibility of this framework using independent datasets and explore its adaptability for other cancers or geographic contexts.”

4.The conclusion suggests the framework can be extended to other cancer types, but the study exclusively analyzes lung cancer in a single region.

Response: We agree that this study focuses solely on lung cancer within Queensland. This was a deliberate decision to allow sufficient depth in describing the methodological framework and to interpret the epidemiological findings in detail for a single cancer type. Our intention was to first establish and validate the feasibility of the REWIRED framework before applying it to other cancers or regions.

We have clarified this point in the Discussion section and acknowledged this as a limitation:

“Nevertheless, the LcVI has not yet been externally validated, and its generalisability to other settings remains to be confirmed. Future research should test the reproducibility of the REWIRED framework using independent datasets and explore its adaptability for other cancers or geographic contexts.”

Reviewer #3: The manuscript introduces a novel methodological framework (REWIRED) for generating a Lung Cancer Vulnerability Index (LcVI) using ecological, registry-based, and socio-environmental data in Queensland, Australia. The approach is technically strong, innovative, and relevant to cancer epidemiology and public health. The paper is generally well written and makes a meaningful contribution.

Overall, the work is sound and the results are convincing. My comments mainly concern interpretation and clarity, and I believe these can be addressed with relatively minor revisions.

Main Comments:

1. Interpretation of diabetes prevalence: The finding that diabetes prevalence was the strongest predictor is interesting. However, the claim that it may serve as a proxy for historical smoking behaviour should be framed more cautiously as a hypothesis, unless further evidence is provided.

Response: Thank you for highlighting this concern. We have revised the Discussion to clarify that the interpretation of diabetes prevalence as a proxy for historical smoking behaviour should be regarded as a hypothesis requiring further evidence. The newly added sentences are shown below in bold:

“Therefore, while our results would need confirmation through other studies, the population prevalence of diabetes could potentially provide a more effective indicator of long-term smoking behaviours among the residents compared to the current smoking prevalence, although this should be interpreted as a hypothesis that requires further empirical validation.”

“If this interpretation is confirmed through further validation, this may increase the rationale for using diabetes prevalence as an ecological measure of previous smoking behaviour, rather than using current smoking prevalence.”

2. Ecological fallacy: The risk of ecological inference should be emphasized more clearly throughout, to ensure readers do not mistakenly extend conclusions to individual-level risk.

Response: Thank you for highlighting this important point.

We have added two sentences at the end of the Abstract to emphasise this:

“As an ecological analysis, the associations reflect relationships at the population level. Future research incorporating individual-level data is needed to confirm whether the area-level associations observed here hold true for individuals.”

In addition, we have added a paragraph in the Discussion to acknowledge the limitation of ecological fallacy:

“As an ecological analysis, this study is subject to the inherent limitation of ecological fallacy. The associations identified between area-level characteristics and lung cancer incidence reflect relationships at the population or geographic level and cannot be assumed to apply to individuals within those areas. For example, areas with higher diabetes prevalence or lower fruit intake may have higher lung cancer incidence, but this does not imply that individuals with diabetes or inadequate fruit intake necessarily have an increased risk of lung cancer. The purpose of this study was to identify spatial patterns and potential contextual determinants of disease burden, not to infer individual-level causation. Future research incorporating individual-level data is needed to confirm whether the area-level associations observed here hold true for individuals.”

Last, we mentioned ecological fallacy as the last sentence of the main text in Conclusion:

“As this is an ecological study, the findings should be interpreted with caution due to the potential for ecological fallacy.”

3. Validation: The LcVI is validated internally but not externally. Please expand discussion of this limitation and its implications for generalizability.

Response: We acknowledge that the LcVI was validated internally but not externally. This is a recognised limitation due to the absence of comparable external datasets containing small-area lung cancer incidence and matching area-level covariates. Future research using independent data sources or other cancer outcomes will be essential to externally validate and test the generalisability of this framework.

We have added a statement in the Discussion section to clarify these points and acknowledge this limitation:

“Nevertheless, the LcVI has not yet been externally validated, and its generalisability to other settings remains to be confirmed. Future research should test the reproducibility of this framework using independent datasets and explore its adaptability for other cancers or geographic contexts.”

4. Clarity of presentation: Some sections describing the REWIRED method are mathematically dense. A simplified schematic or clearer narrative would help readers unfamiliar with advanced ML methods.

Response: Thank you for this helpful suggestion. Since this is the first paper introducing the REWIRED method, we retained sufficient methodological detail to ensure reproducibility and allow other researchers to apply the approach in future studies. However, to assist readers who may be less familiar with machine-learning concepts, we added a simplified overview of the process, presented in Figure 1, which outlines the key steps in the REWIRED method in a clear and accessible format. This schematic now visually highlights the workflow from data inputs to the generation of the Lung Cancer Vulnerability Index (LcVI).

Minor Points:

1. Figures 2 and 3 would benefit from more descriptive captions.

Response: We added descriptive notes for Figure 2 and 3 as:

“Figure 2 Distribution of the lung cancer vulnerability index (LcVI) by categories of lung cancer standardised incidence ratio (SIR) across Queensland.

Note: The violin plots show the distribution of LcVI values for areas classified as having below-average (blue), average (yellow), or above-average (red) lung cancer incidence using exceedance probabilities. The LcVI ranges from 0 to 10, with higher values indicating greater area-level vulnerability. Areas with below-average SIR had a median LcVI of 2.28 (Q1–Q3: 0.85–3.72), those with average SIR had a median of 5.30 (Q1–Q3: 3.88–6.55), and those with above-average SIR had a median of 8.03 (Q1–Q3: 6.98–9.16). The clear gradient across categories demonstrates that the LcVI effectively differentiates areas by their lung cancer incidence.”

“Figure 3 Relationship between lung cancer vulnerability index (LcVI, upper) or relative socio-economic disadvantage (IRSD, lower) and

---

## [Decision Letter · Decision Letter 1]

29 Jan 2026

Generation of machine-learning derived Cancer Vulnerability Indicator to determine the spatial burden of cancer outcomes

PONE-D-25-05126R1

Dear Dr. Baade,

We’re pleased to inform you that your manuscript has been judged scientifically suitable for publication and will be formally accepted for publication once it meets all outstanding technical requirements.

Kind regards,

Godwin Banafo Akrong, Ph.D.

Academic Editor

PLOS One

Additional Editor Comments (optional):

Reviewers' comments:

Reviewer's Responses to Questions

**Comments to the Author**

Reviewer #1: All comments have been addressed

Reviewer #3: All comments have been addressed

2. Is the manuscript technically sound, and do the data support the conclusions?

Reviewer #1: Yes

Reviewer #3: Yes

3. Has the statistical analysis been performed appropriately and rigorously?

Reviewer #1: Yes

Reviewer #3: Yes

4. Have the authors made all data underlying the findings in their manuscript fully available?

Reviewer #1: Yes

Reviewer #3: Yes

5. Is the manuscript presented in an intelligible fashion and written in standard English?

Reviewer #1: Yes

Reviewer #3: Yes

Reviewer #1: My review comments have been specifically addressed and replied to by the author. This is a valuable academic paper.

Reviewer #3: I have carefully reviewed the revised manuscript and the detailed responses to the reviewers’ and editor’s comments. I am satisfied that all substantive concerns raised in the previous review round have been thoroughly and appropriately addressed.

The authors have clearly clarified that the feature importance weights derived from the random forest model reflect predictive relevance rather than causal inference, which is now well articulated in the Discussion. The interpretation of diabetes prevalence has been appropriately reframed as a hypothesis rather than a causal claim, and the ecological nature of the study and the associated risk of ecological fallacy are now clearly and consistently acknowledged across the abstract, discussion, and conclusion.

The comparison between the lung cancer vulnerability index (LcVI) and the IRSD index has been strengthened with a more balanced and transparent justification, including a clear explanation of why IRSD was used as a reference benchmark and an explicit discussion of the limitations and need for external validation. The conclusion has also been improved to better reflect the study’s contributions, practical implications, and future research directions.

Overall, the manuscript is now clearer, more cautious in interpretation, and methodologically sound. The revisions have substantially improved the clarity, robustness, and transparency of the work. I have no further substantive comments and support the manuscript in its current form.

**Do you want your identity to be public for this peer review?** For information about this choice, including consent withdrawal, please see our Privacy Policy

Reviewer #1: No

Reviewer #3: No

---

## [Editor Report · Acceptance letter]

PONE-D-25-05126R1

PLOS One

Dear Dr. Baade,

I'm pleased to inform you that your manuscript has been deemed suitable for publication in PLOS One. Congratulations! Your manuscript is now being handed over to our production team.

Kind regards,

on behalf of

Dr. Godwin Banafo Akrong

Academic Editor

PLOS One